# Metagenome analysis from the sediment of river Ganga and Yamuna: In search of beneficial microbiome

Bijay Kumar Behera[ID][1]*, Biswanath Patra[1], Hirak Jyoti Chakraborty[1], Parameswar Sahu[1], Ajaya Kumar Rout[1], Dhruba Jyoti Sarkar[1], Pranaya Kumar Parida[1], Rohan Kumar Raman[1], Atmakuri Ramakrishna Rao[2], Anil Rai[3], Basanta Kumar Das[1]*, Joykrushna Jena[2], Trilochan Mohapatra[2]

1 Aquatic Environmental Biotechnology and Nanotechnology Division, ICAR-Central Inland Fisheries Research Institute, Barrackpore, West Bengal, India, 2 Indian Council of Agricultural Research, New Delhi, India, 3 ICAR-Indian Agricultural Statistics Research Institute, New Delhi, India

* beherabk18@yahoo.co.in (BKB); basantakumard@gmail.com (BKD)

## Abstract

Beneficial microbes are all around us and it remains to be seen, whether all diseases and disorders can be prevented or treated with beneficial microbes. In this study, the presence of various beneficial bacteria were identified from the sediments of Indian major Rivers Ganga and Yamuna from nine different sites using a metagenomic approach. The metagenome sequence analysis using the Kaiju Web server revealed the presence of 69 beneficial bacteria. Phylogenetic analysis among these bacterial species revealed that they were highly diverse. Relative abundance analysis of these bacterial species is highly correlated with different pollution levels among the sampling sites. The PCA analysis revealed that *Lactobacillus* spp. group of beneficial bacteria are more associated with sediment sampling sites, KAN-2 and ND-3; whereas *Bacillus* spp. are more associated with sites, FAR-2 and ND-2. This is the first report revealing the richness of beneficial bacteria in the Indian rivers, Ganga and Yamuna. The study might be useful in isolating different important beneficial microorganisms from these river sediments, for possible industrial applications.

## Introduction

Rivers are known to be important for the development of human civilization, culture, and welfare. They are one of the crucial components of freshwater ecosystems, maintaining large biodiversity which is vital for sustenance of the terrestrial biome. Since rivers are significant reservoirs of the microbiome, they are relentlessly being explored for the search of *de novo* microbiota. These bacteria are of greater importance due to their different benefits to humans as well as all other strata of organisms present in the trophic pyramid [1]. It provides its rewarding effects generally through four main mechanisms *i.e.* enhancement of barrier function, intervention with host pathogens, immunomodulation, and assembly of neurotransmitters [2]. These organisms are gaining increasing importance as functional foods as well as

**Data Availability Statement:** The metagenomic sequences used in this study have been submitted to the NCBI-SRA database under Accession Nos: SRP190174, SRP190175, SRP189880,

SRP191076, SRP191079, SRP191075, SRP191073, SRP191080, and SRP191499.

**Funding:** The authors received no specific funding for this work.

**Competing interests:** No authors have competing interests

prophylactic, therapeutic, and growth supplements for humans [3–5]. Some of the most common human gut probiotics *viz*. *Lactobacillus* and *Enterococcus* are reported to counteract diabetes, obesity, autoimmune disorder, and cancer through the production of metabolites like short-chain fatty acids [6]. Not only for humans, nowadays, the important microbiome is also being used in agriculture, including veterinary and fisheries, to benefit the animal physiology by improving their internal and external environment [5, 7, 8]. However, in fisheries, the scope of microbial treatment is enormous and the use of the same is gaining day by day. The latest study on *Labeo rohita* established that dietary administration of a probiotic bacterium, *Bacillus aerophilus* KADR3, improves the disease resistance and enhances the immunity against *Aeromonas hydrophila* infection [9]. Similarly, the dietary application of *B. amyloliquefaciens* CCF7, in *L. rohita*, challenged with a fish pathogenic bacteria, *A. hydrophila* MTCC 1739, showed beneficial effects [10]. Though many reports are present on discovering microbiome from natural streams of other countries, there is very insufficient literature available on the same in context to the Indian subcontinent especially in the large riverine ecosystems like Ganga and Yamuna. Therefore, in the present study, the abundance of different beneficial microbiota in the selected stretches of the river Ganga and Yamuna have been identified through the metagenomics approach. The metagenomics study has overcome the problem of culture-oriented microbiological studies associated with different environmental samples and came out as a potential search tool for detailed screening of supportive microbiome species present in an ecosystem [11]. Since the total DNA extracted from an environmental sample is a snapshot of the entire microbial community, metagenomics analysis makes it easier for a comprehensive evaluation of the native microbial ecology [12]. The recent computational advancement and evolution of next-generation sequencing, which can generate millions of sequences at improved cost and speed, make it possible to detect microbial biodiversity easily and their abundance directly from the environmental samples [13–14]. As per our knowledge, this is the first report, presenting an analysis of a large sediment metagenome dataset from these rivers in search of beneficial bacteria.

## Materials and methods

### Sample collection

A total of nine sediment samples were collected from the river Ganga and Yamuna. From the Ganga river, six sediment samples were collected from different sites *viz*. Ganga Barrage (N $26^0 30.858^{//}$E $80^0 19.114^{//}$) (KAN-1), Jajmau (N $26^0 25.301^{//}$E $80^0 25.282^{//}$) (KAN-2), Jana Village (N $26^0 24.495^{//}$E $80^0 26.904^{//}$) (KAN-3) near Kanpur, Uttar Pradesh, Farakka Barrage (N $24^0 47.804^{//}$E $87^0 55.417^{//}$) (FAR-1), Dhulian (N $24^0 47.804^{//}$E $87^0 55.417^{//}$) (FAR-2), Lalbagh (N $29^0 11.087^{//}$E $88^0 16.079^{//}$) (FAR-3) near Farakka, West Bengal. From the river Yamuna, sediment samples were collected from three different sites *viz*. Wazaribad (N $28^0 42.39^{//}$E $77^0 13.57^{//}$) (ND-1), Okhla barrage (N $28^0 32.51^{//}$E $77^0 18.30^{//}$) (ND-2), Faizupur Khaddar (N $28^0 18.43^{//}$E $77^0 27.52^{//}$) (ND-3) near New Delhi, India (**Fig 1**).

### DNA extraction

The obtained samples from different locations from river Ganga and Yamuna were kept in sterile plastic bags, sealed and transported on ice (4˚C), and afterward stored at -80˚C until further processing. Metagenomic DNA from these sediment samples were extracted using a soil gDNA isolation kit (Nucleospin Soil). After the isolation of metagenomic DNA, the quality was checked in Nanodrop 2000 and Qubit® 3.0 Fluorometer. The metagenomic library was prepared using sufficient amounts of extracted good quality DNA.

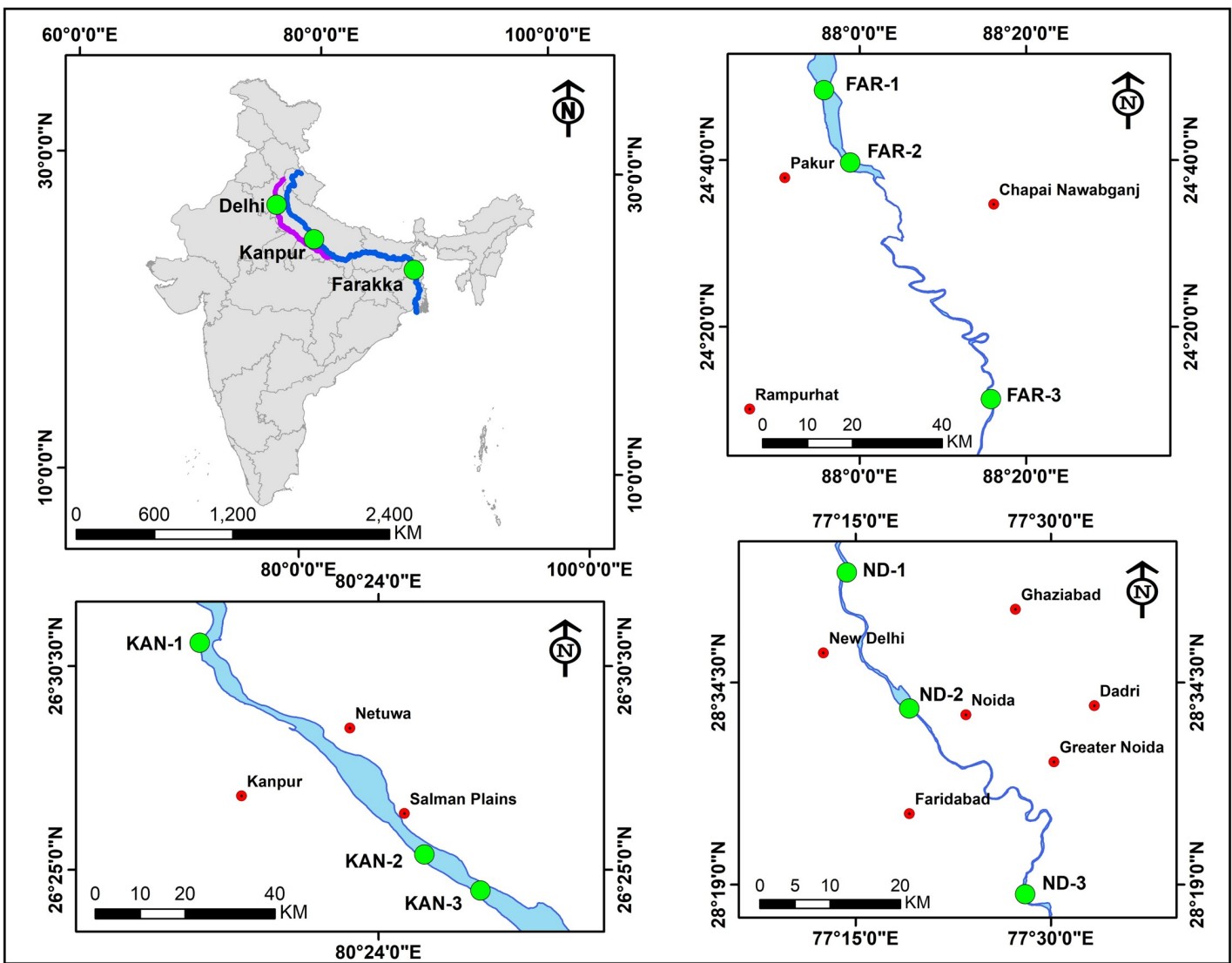

**Fig 1. Map showing the sediment sampling sites.** Sediments were collected from the river Ganga at six locations namely, Kanpur (KAN-1, KAN-2 and KAN-3) and Farakka, (FAR-1, FAR-2 and FAR-3) whereas, three locations from the river Yamuna, at New Delhi (ND-1, ND-2 and ND-3). The map of sediment collection sites was prepared using ArcGIS 10.2.1 platform.

## Metagenomic library preparation

The paired-end sequencing libraries were arranged using Illumina Trueseq Nano DNA Library Prep Kit. Approximately 200ng of eDNA was fragmented by Covaris M220 to produce a mean fragment allocation of 350bp. Covaris shearing produced dsDNA fragments with 3' or 5' over-hangs. The fragments were then subjected to end-repair. As per the description in the kit, the products were PCR amplified with the index primer. The D1000 Screen tape was used to investigate the PCR enriched libraries in the 4200 Tape Station system (Agilent Technologies).

## Whole metagenome sequencing and quality assessment

After obtaining the mean peak size from Agilent Tape Station profile and Qubit concentration for the libraries, PE Illumina libraries were loaded into NextSeq 500 for cluster generation and sequencing. After trimming, a minimum length of 100 nt was applied. The CLC Genomics Workbench

version 8.5.1 (CLC bio; https://www.qiagenbioinformatics.com/products/clc-genomics-workbench) was used to assemble the filtered high-quality reads of each sample into scaffolds.

## Metagenomic sequences accession numbers

The Metagenomic sequences used in this study have been submitted to the NCBI-SRA database under Accession Nos: SRP190174, SRP190175, SRP189880, SRP191076, SRP191079, SRP191075, SRP191073, SRP191080, and SRP191499 for three Kanpur samples (KAN-1, KAN-2, KAN-3), three Farakka samples (FAR-1, FAR-2, FAR-3) and three New Delhi samples (ND-1, ND-2, ND-3) respectively.

## Sequence annotation and bioinformatics analysis

For the detection of the beneficial microbiome in the sediment metagenome, filtered metagenomic reads were used for taxonomical binning by the Kaiju web interface. Classifier-Kaiju used Burrows-Wheeler transform algorithm for taxonomic classification on the protein-level [15]. On the other hand, to highlight the phylogenetic relationship among the beneficial microbiome species found in the sediment metagenome, multiple sequence analysis was carried out using MEGA 6 software. The Neighbor-Joining method was used to infer evolutionary history [16]. The Maximum Composite Likelihood method [17] was used to compute evolutionary distances. To understand the evolutionary relationship among the 69 identified beneficial microbial species, derived from the sediments of the rivers, Ganga and Yamuna, a multiple sequence analysis (MSA) was carried out using MEGA 6 software [18]. Relative abundance of beneficial bacteria was calculated using Kaiju Web Server. Comparison was done based on standard student t-test [19]. Heat map presentation was arranged using multiple experiment viewer (MeV), a standalone tool for visualizing the clustering of multivariate data [20]. The Principal Component Analysis (PCA) biplot and Scatterplot matrix along with correlation values between sampling sites and relative abundance of helpful bacteria were developed in JMP Pro 10 after the standardization of the estimated data.

# Results

## Sequence generation

Sediment samples from nine sites (Fig 1) of river Ganga (KAN-1, KAN-2, and KAN-3; FAR-1, FAR-2, and FAR-3) and river Yamuna (ND-1, ND-2, and ND-3) were analyzed using high throughput next-generation sequencing to identify the microbial biodiversity. The total number of high quality reads with their consequent data volume of each sediment samples are presented in Table 1. All the high quality reads obtained from the sediments of different sites

Table 1. Data details of nine sediment metagenome of river Ganga and Yamuna.

| Location | Description | High quality reads (bp) |
|---|---|---|
| | KAN-1 | 28,718,955 |
| Kanpur | KAN-2 | 33,703,138 |
| | KAN-3 | 33,887,572 |
| | FAR-1 | 24,929,338 |
| Farakka | FAR-2 | 29,128,182 |
| | FAR-3 | 54,496,302 |
| | ND-1 | 64,876,611 |
| New Delhi | ND-2 | 64,749,798 |
| | ND-3 | 62,670,420 |

were considered for sensitive taxonomic classification analysis. However, the taxonomic classification study could not classify all the reads. Only 41.58%, 49.35%, and 54.79% of the total reads were classified form Kanpur sediment samples (KAN-1, KAN-2, and KAN-3) of river Ganga, respectively. In Farakka sediment samples (FAR-1, FAR-2, and FAR-3) of river Ganga, only 50.82%, 52.08%, and 35.68% of the total reads were classified. Similarly, in New Delhi sediment samples (ND-1, ND-2, and ND-3) of river Yamuna, 53.37%, 38.95%, and 44.82% of the total reads were classified, respectively.

## Taxonomical classification of sediment metagenome

Based on the taxonomical classification, a large number of beneficial bacterial species (Table 2) were identified from the sediment samples of the rivers, Ganga, and Yamuna. Four *Vibrio* (*V. mediterranei, V. fluvialis, V. gazogenes,* and *V. alginolyticus),* nine *Bacillus* (*B. clausii, B. circulans, B. subtilis, B. coagulans, B. cereus, B. megaterium, B. mycoides, B. pumilus,* and *B. licheniformis*), sixteen *Lactobacillus* (*L. curvatus, L. brevis, L. casei, L. acidophilus, L. buchneri, L. crispatus, L. delbrueckii, L. fermentum, L. gasseri, L. helveticus, L. johnsonii, L. paracasei, L. plantarum, L. reuteri, L. rhamnosus* and *L. salivarius*), five *Bifidobacterium* (*B. animalis, B. bifidum, B. longum, B. breve,* and *B. adolescentis*), three *Shewanella* (*S. colwelliana, S. putrefaciens* and *S. xiamenensis*), three *Pediococcus* (*P. pentosaceus, P. Acidilactici* and *P. ethanolidurans*), six *Enterococcus* (*E. durans, E. faecium, E. faecalis E. raffinosus E. hirae* and *E. mundtii*), four *Pseudomonas* (*P. fluorescens, P. chlororaphis, P. stutzeri* and *P. synxantha*), four *Roseobacter* (*R. litoralis, R. denitrificans, R. litoralis* and *R. denitrificans*), four *Oenococcus* (*O. oeni, O. kitaharae, O. alcoholitolerans* and *O. oeni* AWRIB429), two *Carnobacterium* (*C. divergens* and *C. maltaromaticum*), two *Streptococcus* (*S. salivarius* and *S. thermophilus*), two *Vagococcus (V. fluvialis* bH819 and *V. teuberi)* along with one each for, *Aeromonas veronii, Leuconostocme senteroides, Micrococcus luteus, Paenibacillus polymyxa,* and *Lactococcus lactis* were identified from the metagenome.

## Phylogenetic analysis

MSA revealed that the majority of the species showed diversity. Phylogenetic tree analysis delineated that, all the species shaped five different clusters (Fig 2). In the first CLUSTER, *S. thermophilus* and *L. brevis* derived from Yamuna and Farakka sediment samples respectively were found phylogenetically very close to each other with the bootstrap value of 34. In CLUSTER-2, *E. faecium* and *L. johnsonii,* derived from Yamuna and Farakka sediment samples respectively, were found very close to each other with a bootstrap value of 14. Similarly, in CLUSTER-3, *L. fermentum* and *L. helveticus* derived from Kanpur and Yamuna sediment samples respectively, were found phylogenetically related with a high bootstrap value of 71. In CLUSTER-4, *P. pentosaceus* and *B. adolescentis* both derived from Yamuna sediment samples were found close to each other with the bootstrap value of 19. The highest numbers of evolutionary closed beneficial microbiome species were found in CLUSTER-5. *L. gasser* and *B. mycoides* derived from Kanpur and Yamuna sediment samples were found close to each other with a bootstrap value of 54.

## Relative abundance at different sites

In the classified metagenomics data, a total of 69 species of the bacteria from 18 different genera were considered for analysis. Heat map analysis showed a clear distinction in the relative abundance of different bacteria between Kanpur and Farakka sediment samples of river Ganga. Similarly, the prevalence of beneficial bacterial species in the sediment samples of river Yamuna was also different from Kanpur and Farakka stretches of river Ganga (Fig 3).

**Table 2. Relative abundance of beneficial bacteria species identified from the nine sediment metagenome of river Ganga and Yamuna.**

| Name of the Genus | Name of the species | KAN-1 | KAN-2 | KAN-3 | Average | SD | FAR-1 | FAR-2 | FAR-3 | Average | SD | ND-1 | ND-2 | ND-3 | Average | SD |
|---|---|---|---|---|---|---|---|---|---|---|---|---|---|---|---|---|
| Lactobacillus | *L. curvatus* | 0.00008 | 0.0002 | 0.00002 | 1.0E-04 | 9.17E-05 | 0.0001 | 0.0001 | 0.00009 | 9.7E-05 | 5.77E-06 | 0.0001 | 0.0001 | 0.0002 | 1.3E-04 | 5.77E-05 |
| | *L. brevis* | 0.0001 | 0.0003 | 0.0002 | 2.0E-04 | 1.00E-04 | 0.0005 | 0.0003 | 0.0006 | 4.7E-04 | 1.5E-04 | 0.0003 | 0.0003 | 0.0003 | 3.0E-04 | 0 |
| | *L. casei* | 0.0004 | 0.0006 | 0.0002 | 4E-04 | 2.00E-04 | 0.0008 | 0.0005 | 0.0004 | 5.7E-04* | 2.1E-04 | 0.0001 | 0.0001 | 0.0001 | 1.0E-04* | 0 |
| | *L. acidophilus* | 0.00003 | 0.00004 | 0.00002 | 3E-05 | 1.00E-05 | 0.00009 | 0.00003 | 0.00004 | 5.33E-05 | 3.21E-05 | 0.00005 | 0.00004 | 0.00007 | 5.33E-05 | 1.52E-05 |
| | *L. buchneri* | 0.00004 | 0.00009 | 0.00003 | 5.3E-05 | 3.2E-05 | 0.00009 | 0.00004 | 0.00008 | 7.0E-05 | 2.65E-05 | 0.00006 | 0.00008 | 0.00009 | 7.67E-05 | 1.52E-05 |
| | *L. crispatus* | 0.0002 | 0.0004 | 0.0002 | 2.7E-04 | 1.2E-04 | 0.0003 | 0.0003 | 0.0002 | 2.7E-04 | 5.7E-05 | 0.0003 | 0.0002 | 0.0004 | 3.0E-04 | 1.0E-04 |
| | *L. delbrueckii* | 0.0001 | 0.0003 | 0.0001 | 1.7E-04 | 1.2E-04 | 0.0003 | 0.0001 | 0.0001 | 1.7E-04 | 1.1E-04 | 0.0003 | 0.0003 | 0.0006 | 04.0E-04 | 1.7E-04 |
| | *L. fermentum* | 0.0002 | 0.0003 | 0.00007 | 1.9E-04 | 1.2E-04 | 0.0003 | 0.0002 | 0.0002 | 2.3E-04 | 5.77E-05 | 0.0003 | 0.0003 | 0.0004 | 3.0E-04 | 1.0E-04 |
| | *L. gasseri* | 0.0001 | 0.0003 | 0.0001 | 1.7E-04 | 1.2E-04 | 0.0002 | 0.0001 | 0.00006 | 1.2E-04 | 7.21E-05 | 0.0002 | 0.0001 | 0.0003 | 2.0E-04 | 1.0E-04 |
| | *L. helveticus* | 0.00005 | 0.0001 | 0.00005 | 6.67E-05 | 2.88E-05 | 0.0001 | 0.00008 | 0.00006 | 8.0E-05 | 2.0E-05 | 0.0001 | 0.00008 | 0.0002 | 1.3E-04 | 6.4E-05 |
| | *L. johnsonii* | 0.00007 | 0.0001 | 0.00003 | 6.67E-05 | 3.5E-05 | 0.0001 | 0.0001 | 0.00007 | 9.0E-05 | 1.73E-05 | 0.0001 | 0.0001 | 0.0001 | 1.0E-04 | 0 |
| | *L. paracasei* | 0.0001 | 0.0002 | 0.00008 | 1.3E-04 | 6.43E-05 | 0.0002 | 0.0001 | 0.0001 | 1.3E-04 | 5.77E-05 | 0.0001 | 0.0001 | 0.0002 | 1.3E-04 | 5.7E-05 |
| | *L. plantarum* | 0.0003 | 0.0005 | 0.0003 | 3.7E-04 | 1.2E-04 | 0.0004 | 0.0003 | 0.0003 | 3.3E-04 | 5.77E-05 | 0.0006 | 0.0004 | 0.0007 | 5.7E-04 | 1.5E-04 |
| | *L. reuteri* | 0.0003 | 0.0006 | 0.0002 | 3.7E-04 | 2.1E-04 | 0.0004 | 0.0003 | 0.0002 | 3.0E-04 | 1.0E-04 | 0.0006 | 0.0004 | 0.0007 | 5.7E-04 | 1.5E-04 |
| | *L. rhamnosus* | 0.0002 | 0.0004 | 0.0001 | 2.3E-04 | 1.5E-04 | 0.0004 | 0.0002 | 0.0001 | 2.3E-04 | 1.5E-04 | 0.0002 | 0.0002 | 0.0003 | 2.3E-04 | 5.7E-05 |
| | *L. salivarius* | 0.0003 | 0.0006 | 0.0002 | 3.7E-04 | 2.1E-04 | 0.0005 | 0.0003 | 0.0002 | 3.3E-04 | 1.5E-04 | 0.001 | 0.0005 | 0.001 | 8.3E-04 | 2.9E-04 |
| Bacillus | *B. clausii* | 0.001 | 0.001 | 0.0005 | 8.3E-04 | 2.89E-04 | 0.002 | 0.001 | 0.001 | 1.3E-03* | 5.7E-04 | 0.0002 | 0.0004 | 0.0004 | 3.3E-04* | 1.2E-04 |
| | *B. circulans* | 0.0003 | 0.0006 | 0.0002 | 3.7E-04 | 2.08E-04 | 0.0005 | 0.0004 | 0.0003 | 4.0E-04 | 1.0E-04 | 0.0002 | 0.0004 | 0.0003 | 3.0E-04 | 1.0E-04 |
| | *B. subtilis* | 0.0006 | 0.0008 | 0.0004 | 6.0E-04 | 2.00E-04 | 0.001 | 0.0007 | 0.0006 | 7.7E-04 | 2.0E-04 | 0.0003 | 0.0008 | 0.0007 | 6.0E-04 | 2.6E-04 |
| | *B. coagulans* | 0.001 | 0.002 | 0.0006 | 1.2E-03 | 7.21E-04 | 0.002 | 0.001 | 0.001 | 1.3E-03 | 5.7E-04 | 0.001 | 0.001 | 0.001 | 1.0E-03 | 0 |
| | *B. cereus* | 0.003 | 0.004 | 0.002 | 3.0E-03 | 1.00E-03 | 0.006 | 0.003 | 0.007 | 5.3E-03 | 2.1E-03 | 0.003 | 0.004 | 0.004 | 3.7E-03 | 5.8E-04 |
| | *B. megaterium* | 0.0006 | 0.001 | 0.0003 | 6.3E-04 | 3.51E-04 | 0.0009 | 0.0009 | 0.001 | 9.3E-04 | 5.77E-05 | 0.0005 | 0.001 | 0.0008 | 7.7E-04 | 2.5E-04 |
| | *B. mycoides* | 0.0005 | 0.0006 | 0.0007 | 6.0E-04* | 1.00E-04 | 0.0006 | 0.0005 | 0.0004 | 5.0E-04 | 1.0E-04 | 0.0002 | 0.0004 | 0.0004 | 3.3E-04* | 1.2E-04 |
| | *B. pumilus* | 0.0004 | 0.0006 | 0.0002 | 4.0E-04 | 2.00E-04 | 0.0008 | 0.0006 | 0.0004 | 6.0E-04 | 2.0E-04 | 0.0003 | 0.0008 | 0.0007 | 6.0E-04 | 2.6E-04 |
| | *B. licheniformis* | 0.0002 | 0.0002 | 0.0001 | 1.07E-04 | 5.77E-05 | 0.0003 | 0.0002 | 0.0001 | 2.0E-04 | 1.0E-04 | 0.0001 | 0.0002 | 0.0002 | 1.7E-04 | 5.77E-05 |
| Pediococcus | *P. pentosaceus* | 0.00007 | 0.0002 | 0.00006 | 1.1E-04 | 7.81E-05 | 0.0002 | 0.0002 | 0.0001 | 1.7E-04 | 5.77E-05 | 0.0003 | 0.0001 | 0.0002 | 2.0E-04 | 1.0E-04 |
| | *P. acidilactici* | 0.008 | 0.008 | 0.1 | 3.9E-02 | 5.31E-02 | 0.0003 | 0.0003 | 0.0002 | 2.7E-04 | 5.77E-05 | 0.0003 | 0.0002 | 0.0003 | 2.7E-04 | 5.77E-05 |
| | *P. ethanolidurans* | 0.00007 | 0.00008 | 0.00005 | 6.67E-05 | 1.53E-05 | 0.00007 | 0.00007 | 0.00004 | 6.0E-05 | 1.73E-05 | 0.00006 | 0.00006 | 0.00009 | 7.0E-05 | 1.73E-05 |
| Vibrio | *V. mediterranei* | 0.00005 | 0.00004 | 0.00005 | 4.67E-05 | 5.77E-06 | 0.0001 | 0.00004 | 0.00005 | 6.3E-05 | 3.2E-05 | 0.00003 | 0.00005 | 0.00005 | 4.3E-05 | 1.15E-05 |
| | *V. fluvialis* | 0.0002 | 0.0002 | 0.0003 | 2.3E-04 | 5.77E-05 | 0.0002 | 0.0002 | 0.0002 | 2.0E-04 | 0 | 0.0004 | 0.0001 | 0.0002 | 2.3E-04 | 5.77E-05 |
| | *V. gazogenes* | 0 | 0 | 0 | 0.00 | 0.00E+00 | 0 | 0 | 0 | 0.00 | 0.00 | 0.0003 | 0.0002 | 0.0003 | 2.7E-04** | 5.77E-05 |
| | *V. alginolyticus* | 0.0007 | 0.0008 | 0.0009 | 8.0E-04 | 1.00E-04 | 0.0009 | 0.0008 | 0.0005 | 7.3E-04 | 2.1E-04 | 0.001 | 0.0006 | 0.001 | 8.7E-04 | 2.3E-04 |
| Roseobacter | *R. litoralis* | 0.001 | 0.001 | 0.003 | 1.6E-03 | 1.18E-03 | 0.0009 | 0.0006 | 0.0007 | 7.3E-04 | 1.5E-04 | 0.0020 | 0.0004 | 0.0005 | 9.7E-04 | 8.9E-04 |
| | *R. denitrificans* | 0.001 | 0.0009 | 0.003 | 1.6E-03 | 1.18E-03 | 0.0007 | 0.0006 | 0.0006 | 6.3E-04 | 5.77E-05 | 0.0020 | 0.0004 | 0.0004 | 9.3E-04 | 9.2E-04 |
| | *R. litoralis 149* | 0.00002 | 0.00003 | 0.00009 | 4.67E-05 | 3.79E-05 | 0.00002 | 0.00001 | 0.00001 | 1.3E-05 | 5.77E-06 | 0.000008 | 0.00001 | 0.000005 | 7.7E-06 | 2.5E-06 |
| | *R. denitrificans 114* | 0.00006 | 0.00006 | 0.0002 | 1.1E-04 | 8.08E-05 | 0.00003 | 0.00003 | 0.00002 | 2.7E-05 | 5.77E-06 | 0.00008 | 0.00002 | 0.00003 | 4.3E-05 | 3.2E-05 |
| Vagococcus | *V. fluvialis bH819* | 0 | 0 | 0 | 0* | 0.00E+00 | 0 | 0 | 0.000002 | 6.7E-07* | 1.15E-06 | 0.0003 | 0.0004 | 0.0006 | 4.3E-04** | 1.5E-04 |
| | *V. teuberi* | 0.0002 | 0.0005 | 0.0001 | 2.7E-04 | 2.08E-04 | 0 | 0.0003 | 0.0002 | 1.7E-04 | 1.5E-04 | 0.00006 | 0.00005 | 0.0001 | 7.0E-05 | 2.6E-05 |
| Oenococcus | *O. oeni* | 0.0003 | 0.0004 | 0.0002 | 3.0E-04 | 1.00E-04 | 0.0004 | 0.0003 | 0.0002 | 3.0E-04 | 1.0E-04 | 0.0003 | 0.0003 | 0.0004 | 3.3E-04 | 5.77E-05 |
| | *O. kitaharae* | 0.0002 | 0.0003 | 0.0001 | 2.0E-04 | 1.00E-04 | 0.0004 | 0.0002 | 0.0002 | 2.7E-04 | 1.1E-04 | 0.0001 | 0.0002 | 0.0002 | 1.7E-04 | 1.1E-04 |
| | *O. alcoholitolerans* | 0.0001 | 0.0001 | 0.00006 | 8.67E-05 | 2.31E-05 | 0.0002 | 0.0001 | 0.00009 | 1.3E-04 | 6.1E-05 | 0.00008 | 0.0001 | 0.0002 | 1.3E-04 | 6.4E-05 |
| | *O. oeni AWRIB429* | 0.00001 | 0.000006 | 0.000003 | 6.33E-06 | 3.51E-06 | 0.00002 | 0.00001 | 0.000004 | 1.1E-05 | 8.1E-06 | 0 | 0.000003 | 0.000002 | 1.67E-06 | 1.5E-06 |
| | *P. fluorescens* | 0.04 | 0.01 | 0.02 | 2.3E-02 | 1.53E-02 | 0.01 | 0.008 | 0.008 | 8.7E-03 | 1.2E-03 | 0.01 | 0.005 | 0.007 | 7.3E-03 | 2.5E-03 |

*(Continued)*

**Table 2.** (Continued)

| Name of the Genus | Name of the species | KAN-1 | KAN-2 | KAN-3 | Average | SD | FAR-1 | FAR-2 | FAR-3 | Average | SD | ND-1 | ND-2 | ND-3 | Average | SD |
|---|---|---|---|---|---|---|---|---|---|---|---|---|---|---|---|---|
| *Pseudomonas* | *P. chlororaphis* | 0.002 | 0.002 | 0.003 | 2.3E-03 | 5.77E-04 | 0.002 | 0.002 | 0.001 | 1.7E-03 | 5.7E-04 | 0.002 | 0.001 | 0.001 | 1.3E-03 | 5.8E-04 |
| | *P. stutzeri* | 0.008 | 0.02 | 0.02 | 1.6E-02 | 6.93E-03 | 0.008 | 0.006 | 0.006 | 6.7E-03 | 1.2E-03 | 0.02 | 0.005 | 0.009 | 1.13E-02 | 7.8E-03 |
| | *P. synxantha* | 0.0002 | 0.0002 | 0.0003 | 2.3E-04 | 5.77E-05 | 0.0002 | 0.0001 | 0.0001 | 1.3E-04 | 5.7E-05 | 0.0003 | 0.0001 | 0.0001 | 1.7E-04 | 1.2E-04 |
| | *S. colwelliana* | 0.0008 | 0.0006 | 0.0009 | 7.7E-04* | 1.53E-04 | 0.0007 | 0.0005 | 0.0005 | 5.7E-04 | 1.2E-04 | 0.0005 | 0.0004 | 0.0005 | 4.7E-04* | 5.77E-05 |
| *Shewanella* | *S. putrefaciens* | 0.0003 | 0.0007 | 0.006 | 2.3E-03 | 3.18E-03 | 0.0003 | 0.0002 | 0.0002 | 2.3E-04 | 5.77E-05 | 0.0004 | 0.0002 | 0.0003 | 3.0E-04 | 1.0E-04 |
| | *S. xiamenensis* | 0.0001 | 0.0003 | 0.0004 | 2.6-E04 | 1.53E-04 | 0.0002 | 0.0001 | 0.0001 | 1.3E-04 | 5.77E-05 | 0.0004 | 0.0001 | 0.0002 | 2.3E-04 | 1.5E-04 |
| | *E. durans* | 0.00009 | 0.0002 | 0.00005 | 1.1E-04 | 7.77E-05 | 0.0001 | 0.0001 | 0.0001 | 1.0E-04 | 0 | 0.0001 | 0.0002 | 0.0003 | 2.0E-04 | 1.0E-04 |
| | *E. faecium* | 0.0003 | 0.001 | 0.0002 | 5.0E-04 | 4.36E-04 | 0.0006 | 0.0003 | 0.0004 | 4.3E-04* | 1.5E-04 | 0.001 | 0.0006 | 0.001 | 8.7E-04* | 2.3E-04 |
| *Enterococcus* | *E. faecalis* | 0.0006 | 0.002 | 0.0005 | 1.0E-03 | 8.39E-04 | 0.0008 | 0.0006 | 0.0005 | 6.3E-04 | 1.5E-04 | 0.001 | 0.0009 | 0.002 | 1.3E-03 | 6.1E-04 |
| | *E. raffinosus* | 0.00001 | 0.00006 | 0.00001 | 2.67E-05 | 2.89E-05 | 0.00002 | 0.00001 | 0.00002 | 1.7E-05 | 5.77E-06 | 0.00004 | 0.00002 | 0.00005 | 3.67E-05 | 1.5E-05 |
| | *E. hirae* | 0.00002 | 0.00009 | 0.00001 | 4.0E-05 | 4.36E-05 | 0.00007 | 0.00003 | 0.00003 | 4.3E-05 | 2.3E-05 | 0.0006 | 0.00007 | 0.0001 | 2.7E-04 | 3.0E-04 |
| | *E. mundtii* | 0.0001 | 0.0003 | 0.00009 | 1.6E-04 | 1.18E-04 | 0.0002 | 0.0001 | 0.0001 | 1.3E-04 | 5.77E-05 | 0.0002 | 0.0002 | 0.0002 | 2.0E-04 | 0 |
| | *B. animalis* | 0.0003 | 0.0004 | 0.0002 | 3.0E-04 | 1.00E-04 | 0.0004 | 0.0003 | 0.0002 | 3.0E-04 | 1.0E-04 | 0.0005 | 0.0002 | 0.0003 | 3.3E-04 | 1.5E-04 |
| *Bifidobacterium* | *B. bifidum* | 0.0002 | 0.0004 | 0.0001 | 2.3E-04 | 1.53E-04 | 0.0003 | 0.0002 | 0.0002 | 2.3E-04 | 5.7E-05 | 0.003 | 0.0003 | 0.0008 | 1.4E-03 | 1.4E-03 |
| | *B. longum* | 0.0004 | 0.002 | 0.0004 | 9.3E-04 | 9.24E-04 | 0.0004 | 0.0004 | 0.0004 | 4.0E-04 | 0 | 0.01 | 0.001 | 0.003 | 4.7E-03 | 4.7E-03 |
| | *B. breve* | 0.0002 | 0.0005 | 0.0002 | 3.0E-04 | 1.73E-04 | 0.0004 | 0.0002 | 0.0002 | 2.7E-04 | 1.2E-04 | 0.002 | 0.0003 | 0.0008 | 1.0E-03 | 8.7E-04 |
| | *B. adolescentis* | 0.0004 | 0.004 | 0.0007 | 2.0E-03 | 2.0E-03 | 0.0004 | 0.0003 | 0.0003 | 3.0E-04 | 1.0E-04 | 0.03 | 0.002 | 0.009 | 1.4E-02 | 1.5E-02 |
| Carnobacterium | *C. divergens* | 0.0003 | 0.0006 | 0.0002 | 3.7E-04 | 2.1E-04 | 0.0003 | 0.0003 | 0.0002 | 2.7E-04 | 5.77E-05 | 0.0002 | 0.0003 | 0.0004 | 3.0E-04 | 1.0E-04 |
| | *C. maltaromaticum* | 0.0004 | 0.001 | 0.0003 | 5.7E-04 | 3.8E-04 | 0.0006 | 0.0004 | 0.0004 | 4.7E-04 | 1.2E-04 | 0.0005 | 0.0004 | 0.0007 | 5.0E-04 | 1.5E-04 |
| Lactococcus | *L. lactis* | 0.0007 | 0.002 | 0.0004 | 1.03E-03 | 8.5E-04 | 0.001 | 0.0008 | 0.0006 | 8.0E-04 | 2.0E-04 | 0.001 | 0.0009 | 0.002 | 1.3E-03 | 6.0E-04 |
| Leuconostock | *L. mesenteroides* | 0.0002 | 0.0005 | 0.002 | 9.0E-04 | 9.6E-04 | 0.0003 | 0.0002 | 0.0002 | 2.3E-04 | 5.77E-05 | 0.0007 | 0.0003 | 0.001 | 6.7E-04 | 3.5E-04 |
| Micrococcus | *M. luteus* | 0.0007 | 0.003 | 0.0009 | 1.5E-03 | 1.3E-03 | 0.002 | 0.001 | 0.0008 | 1.3E-03 | 6.4E-04 | 0.001 | 0.0004 | 0.0006 | 6.7E-04 | 3.0E-04 |
| Streptococcus | *S. salivarius* | 0.00007 | 0.0003 | 0.00002 | 1.3E-04 | 1.5E-04 | 0.00009 | 0.00004 | 0.00005 | 6.0E-05 | 2.64E-05 | 0.0004 | 0.0001 | 0.0004 | 3.0E-04 | 1.7E-04 |
| | *S. thermophilus* | 0.00009 | 0.0003 | 0.00005 | 1.5E-04 | 1.3E-04 | 0.0001 | 0.00008 | 0.00007 | 8.3E-05** | 1.5E-05 | 0.0003 | 0.0003 | 0.0004 | 3.0E-04** | 5.77E-05 |
| Paenibacillus | *P. polymyxa* | 0.0009 | 0.001 | 0.0005 | 8E-04 | 2.6E-04 | 0.001 | 0.001 | 0.0008 | 9.0E-04 | 1.2E-04 | 0.0008 | 0.001 | 0.001 | 9.3E-04 | 1.2E-04 |
| Aeromonas | *A. veronii* | 0.001 | 0.002 | 0.004 | 2.3E-03 | 1.5E-03 | 0.0008 | 0.0006 | 0.0006 | 6.7E-04 | 1.2E-04 | 0.002 | 0.0006 | 0.0009 | 1.2E-03 | 7.3E-04 |

$p \leq 0.05$ * $p \leq 0.01$ ** denote the level significance in Student t-test among average value in the respective row.

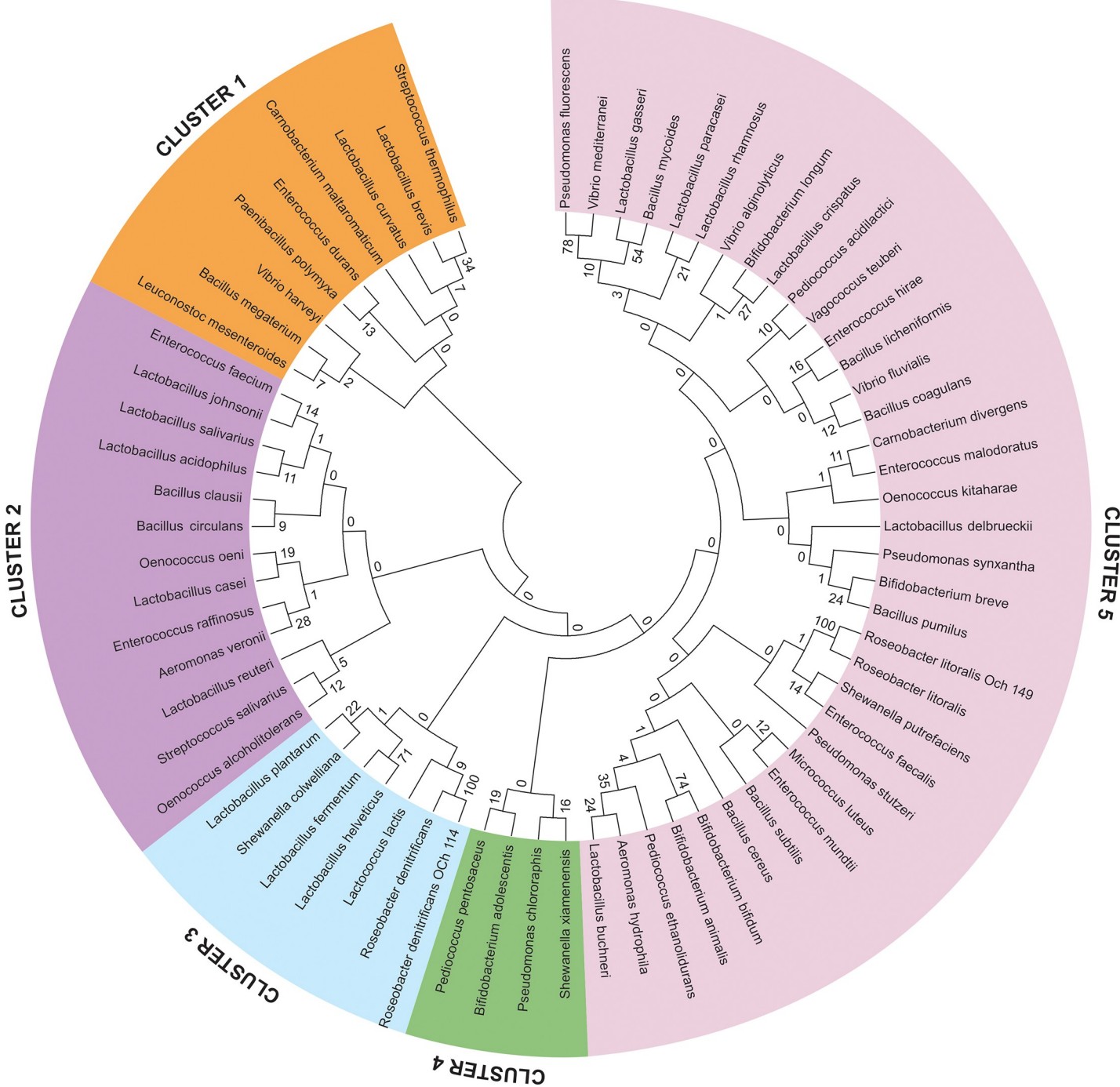

**Fig 2. Phylogenetic tree of 69 identified genome of helpful bacterial species derived from the sediments metagenome of the river Ganga and Yamuna.**

Relative abundance analysis revealed that the species *L. curvetus* and *L. brevis* were present in similar proportion in sediment samples of all the nine sampling sites of the two rivers; however, *L. casei* was present in relatively high proportion at Farakka stretch of river Ganga with statistical significance (p-value of 0.02). *B. clausii* was found in a high proportion ($p \leq 0.05$) at Farakka stretch whereas, *B. mycoides* found in a high proportion ($p \leq 0.05$) at Kanpur stretch of

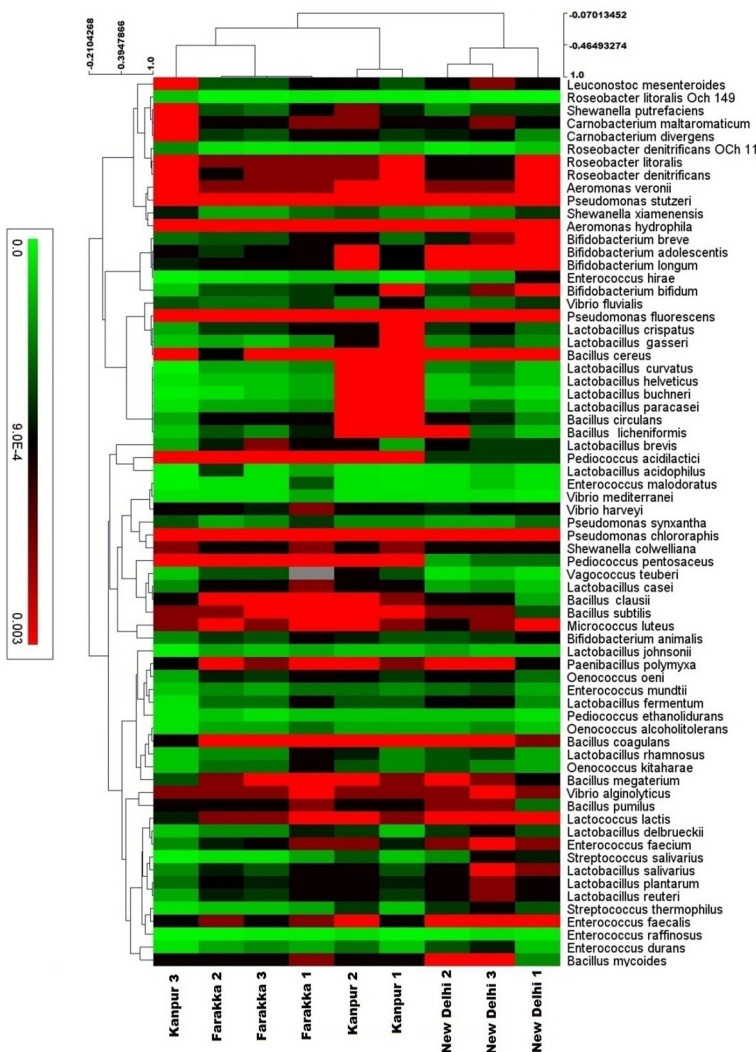

**Fig 3. Heat map of relative species abundance of identified beneficial bacteria from nine different sediment metagenome.** Heat map showing 69 species of beneficial bacteria with significant differences of relative abundances among the nine sampling sites of river Ganga and Yamuna.

river Ganga. Our metagenomic data showed that, one species of *Vibrio* (*V. harveyi*) which showed differential relative abundance between three locations (Kanpur, Farakka and New Delhi) and was found relatively lower (p≤0.05) proportion at New Delhi stretch of river Yamuna as compared to Kanpur stretch of river Ganga. Similarly, *S. colwelliana* was found in a higher proportion (p≤0.05) at Kanpur stretch of river Ganga. *E. faecium* was found in high proportion at New Delhi stretch of river Yamuna as compared to other locations (p≤0.05) (**Table 2**).

Based on the taxonomical hierarchy, it was revealed that, in all the three locations (Kanpur, Farakka, and New Delhi), *L. curvatus* had similar relative abundances. The species, *L. brevis* also showed a similar trend, however, its relative abundance was comparatively higher in the sediment samples of Farakka stretch of river Ganga. The *L. casei* showed lower abundance in sediment samples at New Delhi stretch of river Yamuna as compared to the other two sites of river Ganga (**Fig 4A**). Among the *Pediococcus* population, it is interesting to note that, in the sediment metagenome of Kanpur site of river Ganga, the *P. acidilactici* was (Student's t-test,

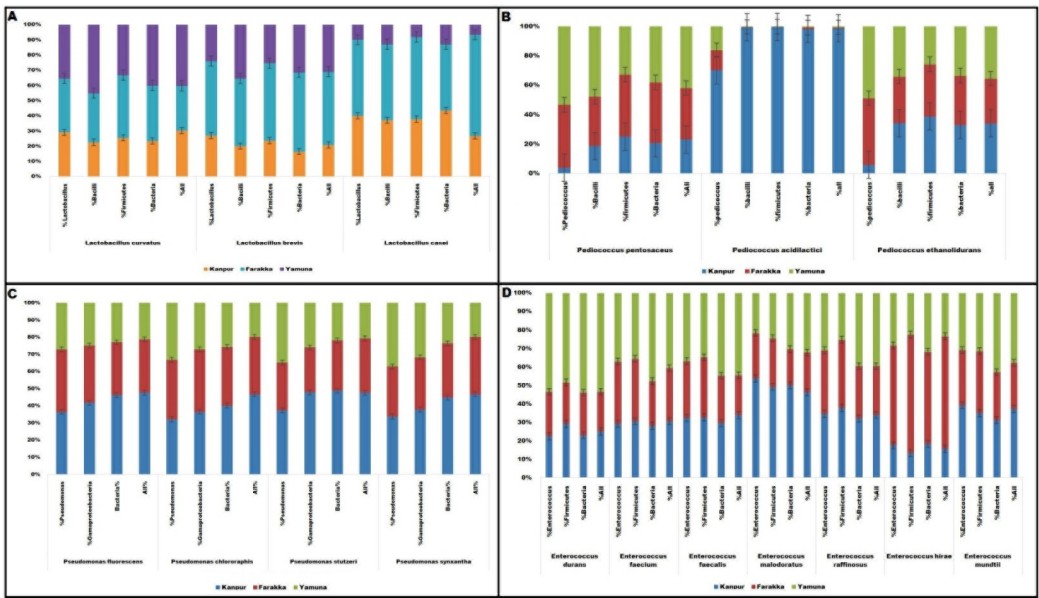

**Fig 4. Relative abundance of beneficial microbes on the basis of their taxonomical profile, where A, B, C, and D represent *Lactobacillus*, *Prediococcus*, *Pseudomonas* and *Enterococcus* groups of different beneficial microbes respectively.** (**A**) *L. casei* showed significantly (Student's t-test, p ≤0.05) lower abundance in New Delhi as compared to rest two locations. (**B**) *Prediococcus pentosaceus* and *Prediococcus ethanolidurans* showed equal abundance among the Farakka and New Delhi locations. *Prediococcus acidilacticii*is significantly dominant over the all taxonomical profile at Kanpur (Student's t-test, p ≤0.05). (**C**) Found significant difference (Student's t-test, p ≤0.05) for *Pseudomonas fluorescens*, *Pseudomonas chlororaphis*. (**D**) *Enterococcus faecium* and *Enterococcus faecalis* showed higher abundance (Student's t-test, p ≤0.05) in Kanpur and Farakka, respectively.

p ≤0.05) dominant over the all taxonomical profile; however, *P. pentosaceus* and *P. ethanoli-durans* showed equal distribution among the sediment metagenomes at Farakka of river Ganga and New Delhi of river Yamuna (**Fig 4B**). Likewise, *Pseudomonas* population showed an equal distribution of relative abundance in all the nine sites. However, *P. fluorescens*, *P. chlororaphis* showed (Student's t-test, p ≤0.05) relative abundance value at Kanpur (**Fig 4C**). Among the *Enterococcus* spp., *E. durans*, *E. malodoratus*, *E. raffinosus*, *E. hirae*, and *E. mundtii* showed non-significant differences among the nine sampling sites. *E. faecium* and *E. faecalis* showed higher abundance (Student's t-test, p ≤0.05) in sediment metagenomes of river Yamuna compared to Kanpur and Farakka stretch of river Ganga (**Fig 4D**).

The biplot of principal component analysis (PCA), the PC1, and PC2 altogether could explain 64% variability in the data which showed that the sites at Farraka are closely associated and sites at Kanpur and New Delhi are diverse about the relative abundance of beneficial bacteria (**Fig 5A**). The relative abundance of beneficial bacteria is found to be closely associated at site FAR-1, KAN-2, KAN-3, ND-1, ND-2, and ND-3. Further, PCA showed that *Lactobacillus* spp. group of beneficial bacteria are more associated with sites KAN-2 and ND-3; whereas *Bacillus* spp. are more associated with FAR-2 and ND-2. The Scatter plot matrix showed the correlation between the sites about the relative abundance of beneficial bacteria (**Fig 5B**). Highest positive correlation was found between ND-2 and ND-3 (r = 0.48) followed by FAR-1 and FAR-2 (r = 0.36) and KAN-1 and KAN-3 (r = 0.33).

## Discussion

The study found that, the river Ganga and Yamuna host several beneficial bacterial genera with enormous taxonomical diversities. Altogether the study could identify 69 beneficial

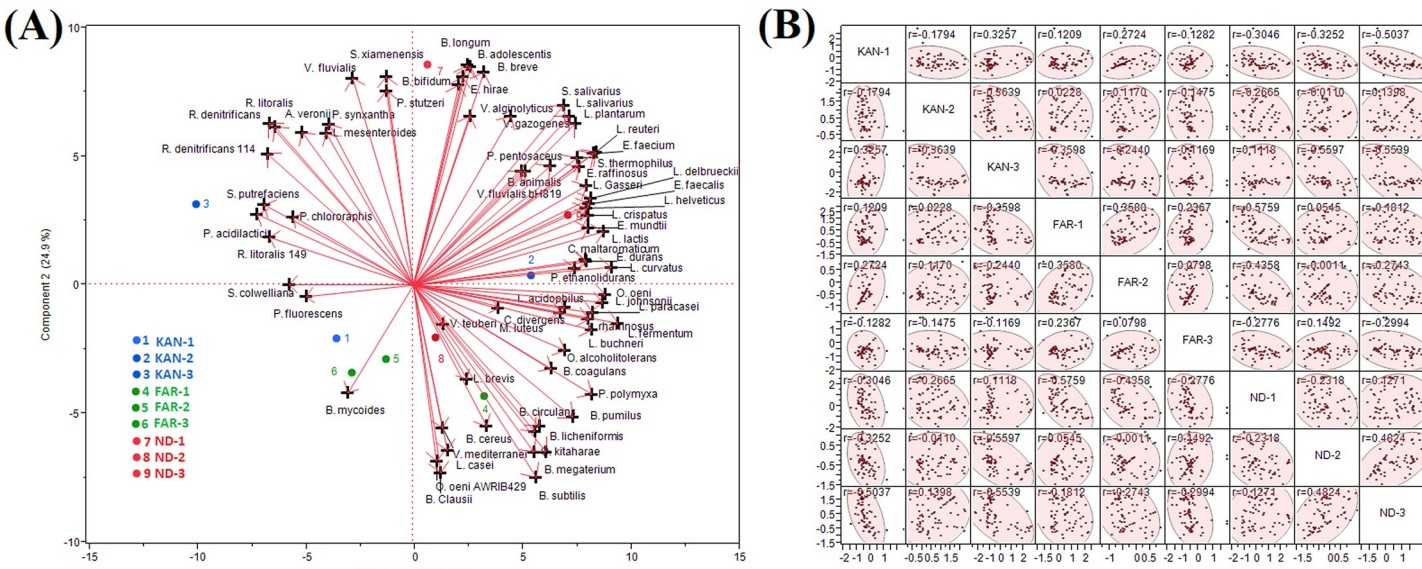

**Fig 5. Biplot of identified helpful bacterial species using Principal Component Analysis (PCA) between two principal component PC1 and PC2 of the river Ganga and Yamuna.**

species belonging to 18 genera (**Table 2**). All the identified beneficial bacteria with their proposed mechanism of action are represented in **S1 and S2 Tables**. The bacterial communities and their functional genomics in sediments and water of the Apies River, South Africa were analyzed using Metagenomic data. Higher diversity in the microbial species associated with the different land uses in the water and sediments of the Apies River was revealed in this study [21]. The taxonomic classification was also previously used to classify microbe strains with consistent categorization at the species level with appropriate safety evaluation, quality assurance, and non-fraudulent labeling [15, 22–26]. In the present study, the beneficial bacterial species under genus *Lactobacillus* (*L. curvatus*, *L. brevis*, *L. helveticus*, *L. gasseri*, *L. crisptus* and *L. casei*, etc.) were identified. These *Lactobacillus* species were reported to exert their beneficial effects by reducing soreness in inflammatory bowel disease (IBD) by producing anti-inflammatory cytokine [27], antibiotic and bacteriostatic activity by the production of bacteriocins [28], and anti-stress activity by the production of β-galactosidase enzymes [29]. *L. curvatus*, was reported to lower the cholesterol level through enhancement of esterase, lipase, cysteine arylamidase, and β-galactosidase activities in the host organisms [30]. *Vibrio* spp. were reported to cause health benefits to the host organism by improving disease resistance through the production of bacteriocin-like substance [31], alteration in the hepatosomatic index, and haemocytes number [32]. *Bacillus* spp. found in the present study were reported to enhance growth, survivability and disease resistance of *Labeo rohita*, and *Macrobrachium rosenbergii* etc. through increased alkaline phosphatase activity, globulin content and lysozyme level [33], enhancement of serum lysozyme activity and serum IgM level [34], increased LYZ gene expression [35], etc. The identified *Bifidobacterium* spp., (*B. animalis*, *B. bifidum*, *B. longum*, *B. breve* and *B. adolescentis*, *etc.*) were reported to attenuate autoimmune encephalomyelitis by inhibiting mononuclear infiltration into the central nervous system [36], diminish gastrointestinal distress by stimulating the production of gastric mucin and other gastrointestinal or neuropeptide hormones [37], anti-obese activities by inhibition of lipid deposit in the liver and adipose tissues [38], alleviate of high-fat diet-induced colitis by inhibition of NF-κB activation and lipopolysaccharide production by gut microbiota [39]. Similarly, *Pediococcus* spp. was

reported to cause many health benefits *viz. P. acidilactici* was reported to advance reproductive performance [40], *P. pentosaceus* has anti-inflammation and anti-cancer effects through mitigation of azoxymethane-induced toxicity [41], *P. ethanol idurans* enhances health through the production of high levels of cellular antioxidant and amplified bile salt hydrolase activities [42]. The identified *Enterococcus faecalis*, was reported to enhance anti-oxidative activity and anti-tumor activity by NK cells and TNF-α [42]. *E. raffinosus* which was reported to prevent bacterial infection in *Labeo rohita* and *Labeo catla* from *E. coli*, *A. hydrophilla*, *S. aerous*, *S. typhimurium* [43]. The identified *E. hirae*, reported producing lipase and bile salt hydrolase enzyme with antioxidant properties, and *E. mundtii* reported with antimicrobial activity [44]. Four *Roseobacter* spp., identified from the sediment metagenomes, were reported with therapeutic value for commercial aquaculture. Earlier, several *Roseobacter* sp. were also reported to reduce fish pathogenic bacteria *V. anguillarum* by *R. clade* [45].

The phylogenetic tree analysis showed that the majority of the species are evolutionary diverse. The phylogenetic tree of all the identified beneficial bacteria species was shaped in five different clusters. In CLUSTER-3, *L. fermentum* and *L. helveticus* derived from Kanpur and Yamuna sediment samples respectively were found phylogenetically related with a high bootstrap value of 71. A similar observation was reported from *Lactobacillus* spp. isolated from animal faeces and it was found that, *L. salivarius* phylogenetic group was closely related to *L. animalis*, *L. apodemi*, and *L. Murinus* [46]. The present finding could be corroborated with a previous report where *Lactococcus* and *Streptococcus* appeared to be closely related and *Lactobacillus* was found to be phylogenetically diverse [27]. The intermixing of phylogenetic distribution, as observed from our study, was also reported previously where *Lactobacillus* and *Pediococcus* were phylogenetically intermixed with 5 species of *Pediococcus* [47]. The *Lactobacillus* chromosomes also expressed the high heterogeneity at phylogenetic, phenotypic, and ecological levels amid the different members of this genus [48]. The present study also found heterogeneity of clustering in *Lactobacillus* species and other beneficial bacteria.

Relative abundance study showed that beneficial bacteria species of different genera were variedly distributed among the three locations; few species are highly dominant in one location over others, *viz. Pediococcus acidilactici* was highly abundant in Kanpur location of river Ganga as compared to other locations. The PCA analysis also showed that the sites at Farraka are closely associated and sites at Kanpur and New Delhi are diverse about the relative abundance of beneficial bacteria. This location-specific change of microbial diversity in the river sediments might be due to differential physiochemical properties and pollution level of the collected sediments. The primary reason for this difference might be due to the release of heavy organic loads and toxic substances (heavy metals, hazardous chemicals, etc.) in some of the selected locations (Kanpur and New Delhi) of these riverine ecosystems through the release of untreated sewage and industrial wastes. It was reported that Kanpur stretch of river Ganga is highly polluted by the untreated effluents from hundreds of tannery industries present in the river bank [49–50]. Very high quantities of diverse heavy metals like Cr, Cu, Pb, Ni, Zn, etc. were found extensively in the water and in the sediments of river Ganga in Kanpur, where pesticide residue like α-HCH, γ-HCH, Dieldrin and Malathion were also reported with a concentration range from 0.190±0.02 to 2.61±0.05 μg/L$^2$ [51]. However, the Farakka stretch of the river Ganga was reported to be less polluted [52]. Like Kanpur stretch of river Ganga, the New Delhi stretch of the river Yamuna was also reported to be severely polluted by heavy metal pollutions due to the release of untreated metropolitan swages, factory effluents, etc. [53, 54]. Therefore, we presume it might be the reason for differences in the relative abundance of beneficial bacteria species among different locations in the river Ganga and Yamuna. Our results could be supported by the previous finding, where the proportion of beneficial microbes in the gastrointestinal microbiota of *Bufo raddei* was altered due to heavy-metal pollution [55]. This

is the first report on the identification of beneficial bacteria in the sediments of the river Ganga and Yamuna, using a metagenomic approach. This study revealed extensive insights on the abundance of native important beneficial microorganisms in these rivers and their functional properties.

## Conclusion

Our research indicates that the sediment metagenome of the river Ganga and Yamuna manifests the enriched microbial distribution of beneficial bacteria. The phylogenetic study of identified useful microbial species revealed that the majority of the species are evolutionarily diverse. This study also refers to the clear distinction in the relative abundance of different beneficial bacteria across the sampling sites. Isolation of different beneficial bacteria from these riverine ecosystems would be highly useful for industrial applications in the future.

## Supporting information

**S1 Table. Health benefit of identified bacteria and their proposed mechanism of action.**
(DOCX)

**S2 Table. Relative abundance of beneficial bacteria species identified from the nine sediment metagenome of river Ganga and Yamuna.**
(DOCX)

## Acknowledgments

Authors are thankful to Mr. Asim Kumar Jana, Senior Technical Assistant, ICAR-CIFRI, Barrackpore, Kolkata, India for sampling and technical assistance. This work has been carried under CABin Scheme.

## Author Contributions

**Conceptualization:** Bijay Kumar Behera, Atmakuri Ramakrishna Rao, Anil Rai, Trilochan Mohapatra.

**Data curation:** Biswanath Patra, Hirak Jyoti Chakraborty, Ajaya Kumar Rout.

**Formal analysis:** Hirak Jyoti Chakraborty, Parameswar Sahu, Ajaya Kumar Rout, Dhruba Jyoti Sarkar, Rohan Kumar Raman.

**Funding acquisition:** Atmakuri Ramakrishna Rao, Anil Rai.

**Investigation:** Bijay Kumar Behera, Basanta Kumar Das, Trilochan Mohapatra.

**Methodology:** Parameswar Sahu, Ajaya Kumar Rout, Pranaya Kumar Parida.

**Project administration:** Bijay Kumar Behera, Atmakuri Ramakrishna Rao, Anil Rai.

**Resources:** Bijay Kumar Behera, Basanta Kumar Das, Joykrushna Jena, Trilochan Mohapatra.

**Software:** Biswanath Patra, Hirak Jyoti Chakraborty, Parameswar Sahu, Ajaya Kumar Rout, Dhruba Jyoti Sarkar, Rohan Kumar Raman.

**Supervision:** Bijay Kumar Behera, Joykrushna Jena.

**Validation:** Biswanath Patra, Parameswar Sahu, Ajaya Kumar Rout, Pranaya Kumar Parida, Rohan Kumar Raman.

**Writing – original draft:** Biswanath Patra, Hirak Jyoti Chakraborty, Ajaya Kumar Rout, Dhruba Jyoti Sarkar, Pranaya Kumar Parida, Rohan Kumar Raman.

**Writing – review & editing:** Bijay Kumar Behera, Atmakuri Ramakrishna Rao, Anil Rai, Basanta Kumar Das, Joykrushna Jena, Trilochan Mohapatra.

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
