## [Decision Letter · Decision Letter 0]

9 Jun 2020

PONE-D-20-11686

Metagenome analysis from the sediment of river Ganga and Yamuna: In search of health beneficial microbiome

PLOS ONE

Dear Dr. Behera,

Thank you for submitting your manuscript to PLOS ONE. After careful consideration, we feel that it has merit but does not fully meet PLOS ONE’s publication criteria as it currently stands. Therefore, we invite you to submit a revised version of the manuscript that addresses the points raised during the review process.

We look forward to receiving your revised manuscript.

Kind regards,

Manas Ranjan Dikhit

Academic Editor

PLOS ONE

Journal Requirements:

2. We note that Figure 1 in your submission contain map/satellite images which may be copyrighted. All PLOS content is published under the Creative Commons Attribution License (CC BY 4.0), which means that the manuscript, images, and Supporting Information files will be freely available online, and any third party is permitted to access, download, copy, distribute, and use these materials in any way, even commercially, with proper attribution. For these reasons, we cannot publish previously copyrighted maps or satellite images created using proprietary data, such as Google software (Google Maps, Street View, and Earth). For more information, see our copyright guidelines: http://journals.plos.org/plosone/s/licenses-and-copyright.

2.1.    You may seek permission from the original copyright holder of Figure 1 to publish the content specifically under the CC BY 4.0 license.

2.2.    If you are unable to obtain permission from the original copyright holder to publish these figures under the CC BY 4.0 license or if the copyright holder’s requirements are incompatible with the CC BY 4.0 license, please either i) remove the figure or ii) supply a replacement figure that complies with the CC BY 4.0 license. Please check copyright information on all replacement figures and update the figure caption with source information. If applicable, please specify in the figure caption text when a figure is similar but not identical to the original image and is therefore for illustrative purposes only.

"No."

Reviewers' comments:

Reviewer's Responses to Questions

**Comments to the Author**

1. Is the manuscript technically sound, and do the data support the conclusions?

Reviewer #1: No

Reviewer #2: Yes

2. Has the statistical analysis been performed appropriately and rigorously? 

Reviewer #1: No

Reviewer #2: Yes

3. Have the authors made all data underlying the findings in their manuscript fully available?

Reviewer #1: Yes

Reviewer #2: Yes

4. Is the manuscript presented in an intelligible fashion and written in standard English?

Reviewer #1: Yes

Reviewer #2: Yes

5. Review Comments to the Author

Reviewer #1: This manuscript by Behera BK et al describes distribution of various beneficial bacterial species in the sediments of two Indian rivers (Ganga and Yamuna) using shutgun metagenomic approach. In this study the authors generated sequencing reads for nine sampling sites; with three sites from three geographically distant regions along the river banks makes the experimental design well thoughtful. They primarily analyzed the sequencing reads using bioinformatics tools and compared average number of species in these three regions. Through system biology based approaches they tried to elucidate the functional relevance of some selected bacteria to aquatic environment.

The study seems to be interesting and well relevant. Although the manuscript depicts to address the problem, it lacks clarity in many aspects and should be addressed before acceptance.

My major concerns are :

1. The criteria used for construction of the beneficial bacteria list is incomprehensible. It looks very strange that beneficial effects of each of these 69 bacteria (Supplementary Table 1) were evidenced by only one study. A simple pubmed search indicates that several of them have pathogenic effect to aquatic organisms. For example, Vibrio mediterranei is associated to major mortality in Pinna nobilis (https://www.sciencedirect.com/science/article/abs/pii/S0044848619324494). Some have role in serious human health hazards. For example consumption of fish infected with V. fluvialis has been reported to cause mild to moderate dehydration, vomiting, fever, abdominal pain and diarrhoea (https://www.ncbi.nlm.nih.gov/pmc/articles/PMC2996184/). There are many more such bacteria from this list which has been reported to be associated with serious illness in human and other aquatic organisms. So authors need to define a clear criteria for this classification and provide more than one evidence supporting their health benefit.

2. The abstract describes that a total of 242 health beneficial bacteria were identified. However, rest of the manuscript discusses only 69 (Heatmap figure legend says 67 species). While Kaiju web server performs taxonomic classification of sequencing reads, it is unclear from the manuscript whether the list of beneficial bacteria was created before performing the all data analysis or bacteria names (form Kaiju results) were selected later after a manual literature search?

3. The method section is ambiguous and insufficient to reproduce the results independently. Important analyses such as relative abundance calculation and statistical methods used for significance tests have not been described sufficiently.

4. In many cases the input data are not clear. For example, the inputs for multiple sequence analysis (or multiple sequence alignment???), species Phylogeny construction and PCA are not clear form the text.

5. The purpose of Phylogenetic analysis should be discussed appropriately. The authors conclude that "some of the microbial species are highly conserved throughout the evolution". How come a species can be conserved? Genes/proteins are conserved during evolution of a lineage.

- Bootstrap is a score to show confidence of relatedness or similarily in a clade and does not refer to species closeness.

- Pg 15: What is "evolutionary closed supportive microbiome"??

- Pg 20: What is the difference between "Circular Cladogram" and "Cladogram"?

6. The Relative abundance calculation is totally unclear. In biodiversity studies usually it refers to the number of individuals from i-th speceis divided by total number of individuals from a sample (10.1590/S1516-89132012000200014). It would be useful if the authors provide a concise definition of this index at species level.

7. The GO enrichment and pathway enrichment does not show any relation with heavy metal re-mediation. Mere presence of HMA domain in one protein from one species should not be concluded as "potential heavy metal remediating property". This result should be reported in a subtle way.

8. The rationale behind the STRING protein-protein interaction is not clear.

9. In introduction, the manuscript describes that there are many studies on discovering microbiome from natural streams of other countries. However, in the discussion the results obtained from present study were not compared enough with results from past metagenomic studies. The discussion should put the results in a more broader context.

10. The conclusion that "sediment metagenome of the river Ganga and Yamuna manifests the enriched microbial distribution of health beneficiary microbes" seems biased. Since the study analyses pre-selected list of 69 species, whether the samples are enriched in beneficial bacteria or not is difficult to conclude.

## Minor Comments

1. Lack of page and line numbers in the manuscript makes it hard to review. I have tried to write the exact phrase so that the authors can find the concerned lines.

2. Reference-1 seems bit out of context. Also the list of authors seems incomplete (Compare with this https://www.scienceopen.com/document?vid=b20de2d9-e8dd-478e-8e2b-5fb9d967b6ea)

3. Pg -12 "of health beneficialmicrobiome" should be "beneficial microbiome"

4. Substitution models and number of bootstrap replicate number should be mentioned in a phylogenetic analysis.

5. For all software tools please provide appropriate citation. Citations help the labs to receive funding and maintain the tool.

6. Pg-12: "each sample were then assembled into scaffolds using CLC". Please provide the number of scaffolds and their average length. There is no information about these scaffolds in the results.

7. Pg 14: Please fix the scientific names: Bacillus Clausii, Enterococcus Hirae, Leuconostocme senteroides, Pediococcus Acidilactici,

Pseudomonas Stutzeri, Roseobacter Litoralis

8. Pg-14: "four Roseobacter (R. litoralis, R. denitrificans, R. Litoralis and R. denitrificans)"- R. litoralis and R. Litoralis are similar or diffenent. It makes the list fof species to 68.

9. Multiple species has Similar genus abbreviation. Consider using two letter abbreviation; for example Vi. mediterranei and Va. fluvialis bH819.

10. "Lactococcuslactis " should be Lactococcus lactis ???

11. "Leuconostocme senteroides" should be Leuconostoc mesenteroides ???

12. "boot strap" or bootstrap ???

13. Pg 15: Heatmap is a way of data representation. What is Heatmap analysis??

14. Pg 15: "beneficialspecies in the" should be "beneficial species in the" ??

15. The term "significantly" should be used more cautiously. It should be used only for statistical tests and the alpha (p-value) must be specified.

16. The term "a large number of" seems very vague. Rather, shuch numbers should be descried in a quantitative manner (in terms of percentage).

17. Pg 35: I would suggest to replace Table 2 with a box-plot. It would reduce the number of pages while making the results more informative and attractive.

18. The bootstrap values should be displayed on the tree in appropriate way.

My intention behind extensive comments should be taken in a constructive manner to improve the manuscript.

Thank you.

Reviewer #2: In the manuscript PONE-D-20-11686 authors have performed metagenome analysis to identify microbes with favorable health outcome to humans as well as all other strata of organisms present in the trophic pyramid. Through contemporary high throughput method authors have reported presence of Lactobacillus, at Farakka stretch of river Ganga and at New Delhi stretches of river Yamuna, whereas Roseobacter spp. was found to be highly enriched at Kanpur sites of river Ganga. Such finding would definitely helpful in isolating beneficial microbes from river sediments for future industrial application. The article meets scientific standard for publication.

However, authors may carry out minor revision to improve quality of the manuscript.

1. In method logic behind setting up string analysis parameter could be mentioned.

2. Authors could mention limitations and future direction of the study in the conclusion.

6. PLOS authors have the option to publish the peer review history of their article (what does this mean?). If published, this will include your full peer review and any attached files.

Reviewer #1: Yes: Kanhu Charan Moharana

Reviewer #2: Yes: Dr. Dibyabhaba Pradhan

---

## [Author Response · Author response to Decision Letter 0]

7 Sep 2020

Response to reviewers

Comment

We note that Figure 1 in your submission contain map/satellite images which may be copyrighted. All PLOS content is published under the Creative Commons Attribution License (CC BY 4.0), which means that the manuscript, images, and Supporting Information files will be freely available online, and any third party is permitted to access, download, copy, distribute, and use these materials in any way, even commercially, with proper attribution. For these reasons, we cannot publish previously copyrighted maps or satellite images created using proprietary data, such as Google software (Google Maps, Street View, and Earth). For more information, see our copyright guidelines:

http://journals.plos.org/plosone/s/licenses-and-copyright.

Thank you for letting us know the Map in Figure 1 was created used ArcGIS 10.2.1 platform.

Before we proceed, please also clarify where the data used in the map is from. If the authors used their own data for the map, please let us know.

Response

Thanks for suggestion.

As per the suggestion, the Figure 1 has been modified using ArcGIS 10.2.1 platform.

The data used in preparation of the map was own data of the authors. 

Reviewer # 1

Comment 1

The criteria used for construction of the beneficial bacteria list are incomprehensible. It looks very strange that beneficial effects of each of these 69 bacteria (Supplementary Table 1) were evidenced by only one study. A simple pubmed search indicates that several of them have pathogenic effect to aquatic organisms. For example, Vibrio mediterranei is associated to major mortality in Pinna nobilis(https://www.sciencedirect.com/science/article/abs/pii/S0044848619324494). Some have role in serious human health hazards. For example consumption of fish infected with V. fluvialis has been reported to cause mild to moderate dehydration, vomiting, fever, abdominal pain and diarrhoea(https://www.ncbi.nlm.nih.gov/pmc/articles/PMC2996184/). There are many more such bacteria from this list which has been reported to be associated with serious illness in human and other aquatic organisms. So authors need to define clear criteria for this classification and provide more than one evidence supporting their health benefit.

Response

The beneficial effects of the identified bacteria from the metagenome sequence data were verified again with more literatures to confirm. Study revealed that, few strains of the identified bacterial species are beneficial in nature; however same bacterial species are also pathogenic which may be due to different strains of same species. The pathogenic activities of the same bacterial species may be due to the variations in genome. They may be representing different ecotype or biotype. The details have been provided as Supplementary Table S1 and S2.

Comment 2

The abstract describes that a total of 242 health beneficial bacteria were identified. However, rest of the manuscript discusses only 69 (Heatmap figure legend says 67 species). While Kaiju web server performs taxonomic classification of sequencing reads, it is unclear from the manuscript whether the list of beneficial bacteria was created before performing the all data analysis or bacteria names (form Kaiju results) were selected later after a manual literature search?

Response

The MS has been modified. The 69 helpful bacteria have been reported in this MS instead of 242. Heat map figure legend has been changed accordingly. The list of bacteria names (form Kaiju results) were selected after a manual literature search.

Comment 3

The method section is ambiguous and insufficient to reproduce the results independently. Important analyses such as relative abundance calculation and statistical methods used for significance tests have not been described sufficiently.

Response

Thanks for suggestion.

The identified beneficial bacteria found in our metagenomic data were manually checked with NCBI references for conformation. Relative abundance of beneficial bacteria was calculated using Kaiju Web Server. Comparison was done based on standard student t-test following the Reference Imchen et al., 2018. They did similar kind of metagenomic research.

Comment 4

In many cases the input data are not clear. For example, the inputs for multiple sequence analysis (or multiple Sequence alignment???), species Phylogeny construction and PCA are not clear from the text.

Response

The Phylogenetic tree has been reconstructed and PCA analysis has been revised.

Comment 5

The purpose of Phylogenetic analysis should be discussed appropriately. The authors conclude that "some of the microbial species are highly conserved throughout the evolution". How come a species can be conserved? Genes/proteins are conserved during evolution of a lineage.

- Bootstrap is a score to show confidence of relatedness or similarity in a clade and does not refer to species closeness.

- Pg 15: What is "evolutionary closed supportive microbiome”??

- Pg 20: What is the difference between "Circular Cladogram" and "Cladogram"?

Response

The Phylogenetic tree has been reconstructed considering 69 identified helpful bacteria with the Bootstrap values. The phylogenetics description part in the MS has been modified accordingly.

Comment 6

The Relative abundance calculation is totally unclear. In biodiversity studies usually it refers to the number of individuals from i-th pecies divided by total number of individuals from a sample (10.1590/S1516-89132012000200014). It would be useful if the authors provide a concise definition of this index at species level.

Response

Thanks for suggestion.

Relative abundance was calculated from total number of bacterial population generated from metagenomic sequences. The MS has been modified as suggested by the Reviewer.

Comment 7

The GO enrichment and pathway enrichment does not show any relation with heavy metal re-mediation. Mere presence of HMA domain in one protein from one species should not be concluded as "potential heavy metal remediating property".

This result should be reported in a subtle way.

Response

The Functional metagenomics analysis in the Result section of the MS has been removed. The same in the Methods and Discussion part of the MS has also been removed.

Comment 8

The rationale behind the STRING protein-protein interaction is not clear.

Response

STRING protein-protein interaction part has been removed from the MS.

Comment 9

In introduction, the manuscript describes that, there are many studies on discovering microbiome from natural streams of other countries. However, in the discussion the results obtained from present study were not compared enough with results from past metagenomic studies. The discussion should put the results in a more broader context.

Response

As suggested by the reviewer, the results obtained from present study has been compared with results from past metagenomic studies in the discussion.

Comment 10

The conclusion that "sediment metagenome of the river Ganga and Yamuna manifests the enriched microbial distribution of health beneficiary microbes" seems biased. Since the study analyses pre-selected list of 69 species, whether the samples are enriched in beneficial bacteria or not is difficult to conclude.

Response

The 69 species of helpful bacteria were identified from our Metagenome data and confirmed with published literatures about their benefits. The detail information has been provided in the Supplementary Table S1 and S2.

Minor Comments 

Comment 1

Lack of page and line numbers in the manuscript makes it hard to review. I have tried to write the exact phrase so that the authors can find the concerned lines.

Response

Modified the MS as suggested

Comment 2

Reference-1 seems bit out of context. Also the list of authors seems incomplete (Compare with this

https://www.scienceopen.com/document?vid=b20de2d9-e8dd-478e-8e2b-5fb9d967b6ea)

Response

Reference-1 has been removed from the MS

Comment 3

Pg -12 "of health beneficial microbiome" should be "beneficial microbiome"

Response

Modified the MS as suggested

Comment 4

Substitution models and number of bootstrap replicate number should be mentioned in a phylogenetic analysis.

Response

The bootstrap replicate number have been mentioned in the phylogenetic tree

Comment 5

For all software tools please provide appropriate citation. Citations help the labs to receive funding and maintain the tool.

Response

Modified the MS as suggested

Comment 6

Pg-12: "each sample was then assembled into scaffolds using CLC". Please provide the number of scaffolds and their Average length. There is no information about these scaffolds in the results.

Response

Modified the MS as suggested

Comment 7

Pg 14: Please fix the scientific names: Bacillus Clausii, Enterococcus Hirae, Leuconostocme senteroides, Pediococcus Acidilactici, Pseudomonas Stutzeri, Roseobacter Litoralis

Response

Corrected the MS as suggested

Comment 8

Pg-14: "four Roseobacter (R. litoralis, R. denitrificans, R. Litoralis and R. denitrificans)"- R. litoralis and R. Litoralis are similar or different. It makes the list of species to 68.

Response

Corrected the MS as suggested

Comment 9

Multiple species has Similar genus abbreviation. Consider using two letter abbreviation; for example Vi. Mediterranei and Va. fluvialis bH819.

Response

Corrected the MS as suggested

Comment 10

"Lactococcuslactis " should be Lactococcus lactis ???

Response

Corrected the MS as suggested

Comment 11

"Leuconostocme senteroides" should be Leuconostoc mesenteroides ???

Response

Corrected the MS as suggested

Comment 12

"boot strap" or bootstrap ???

Response

Corrected the MS as suggested

Comment 13

Pg 15: Heatmap is a way of data representation. What is Heatmap analysis??

Response

Corrected the MS as suggested

Comment 14

Pg 15: "beneficial species in the" should be "beneficial species in the”??

Response

Corrected the MS as suggested

Comment 15

The term "significantly" should be used more cautiously. It should be used only for statistical tests and the alpha (pvalue) must be specified.

Response

Modified the MS as suggested

Comment 16

The term "a large number of" seems very vague. Rather, such numbers should be described in a quantitative manner (in terms of percentage).

Response

Modified the MS as suggested

Comment 17

Pg 35: I would suggest replacing Table 2 with a box-plot. It would reduce the number of pages while making the results more informative and attractive.

Response

Thanks for suggestion

Tabular form in the MS would be more useful

Comment 18

The bootstrap values should be displayed on the tree in appropriate way.

Response

The bootstrap values have been displayed on the tree in appropriate way

Reviewer # 2

Comment 1

In method logic behind setting up string analysis parameter could be mentioned.

Response

The string analysis part has been removed from the MS.

Comment 2

Authors could mention limitations and future direction of the study in the conclusion.

Response

The MS has been modified as suggested

---

## [Editor Report · Decision Letter 1]

10 Sep 2020

Metagenome analysis from the sediment of river Ganga and Yamuna: In search of beneficial microbiome

PONE-D-20-11686R1

Dear Dr. Behera,

We’re pleased to inform you that your manuscript has been judged scientifically suitable for publication and will be formally accepted for publication once it meets all outstanding technical requirements.

Kind regards,

Manas Ranjan Dikhit

Academic Editor

PLOS ONE
---

## [Editor Report · Acceptance letter]

25 Sep 2020

PONE-D-20-11686R1 

Metagenome analysis from the sediment of river Ganga and Yamuna: In search of beneficial microbiome 

Dear Dr. Behera:

I'm pleased to inform you that your manuscript has been deemed suitable for publication in PLOS ONE. Congratulations! Your manuscript is now with our production department. 

Kind regards, 

on behalf of

Dr. Manas Ranjan Dikhit 

Academic Editor

PLOS ONE